# *Hericium erinaceus* in Neurodegenerative Diseases: From Bench to Bedside and Beyond, How Far from the Shoreline?

**DOI:** 10.3390/jof9050551

**Published:** 2023-05-10

**Authors:** Federico Brandalise, Elisa Roda, Daniela Ratto, Lorenzo Goppa, Maria Letizia Gargano, Fortunato Cirlincione, Erica Cecilia Priori, Maria Teresa Venuti, Emanuela Pastorelli, Elena Savino, Paola Rossi

**Affiliations:** 1Department of Biosciences, University of Milan, 20133 Milano, Italy; federico.brandalise@unimi.it; 2Laboratory of Clinical & Experimental Toxicology, Pavia Poison Centre, National Toxicology Information Centre, Toxicology Unit, Istituti Clinici Scientifici Maugeri IRCCS, 27100 Pavia, Italy; elisa.roda@icsmaugeri.it; 3Department of Biology and Biotechnology, University of Pavia, 27100 Pavia, Italy; daniela.ratto@unipv.it (D.R.); ericacecilia.priori@unipv.it (E.C.P.); mariateresa.venuti01@universitadipavia.it (M.T.V.); empas97@icloud.com (E.P.); 4Department of Earth and Environmental Science, University of Pavia, 27100 Pavia, Italy; lorenzo.goppa01@universitadipavia.it (L.G.); elena.savino@unipv.it (E.S.); 5Department of Soil, Plant, and Food Sciences, Via G. Amendola, 165/A, 70126 Bari, Italy; marialetizia.gargano@uniba.it; 6Department of Agricultural, Food and Forest Sciences, University of Palermo, Viale delle Scienze, Bldg. 5, 90128 Palermo, Italy; fortunato.cirlincione@unipa.it

**Keywords:** *Hericium erinaceus*, erinacines, hericenones, ergothioneine, NGF, BDNF, depression, aging, Alzheimer, mild cognitive impairment

## Abstract

A growing number of studies is focusing on the pharmacology and feasibility of bioactive compounds as a novel valuable approach to target a variety of human diseases related to neurological degeneration. Among the group of the so-called medicinal mushrooms (MMs), *Hericium erinaceus* has become one of the most promising candidates. In fact, some of the bioactive compounds extracted from *H*. *erinaceus* have been shown to recover, or at least ameliorate, a wide range of pathological brain conditions such as Alzheimer’s disease, depression, Parkinson’s disease, and spinal cord injury. In a large body of in vitro and in vivo preclinical studies on the central nervous system (CNS), the effects of erinacines have been correlated with a significant increase in the production of neurotrophic factors. Despite the promising outcome of preclinical investigations, only a limited number of clinical trials have been carried out so far in different neurological conditions. In this survey, we summarized the current state of knowledge on *H*. *erinaceus* dietary supplementation and its therapeutic potential in clinical settings. The bulk collected evidence underlies the urgent need to carry out further/wider clinical trials to prove the safety and efficacy of *H*. *erinaceus* supplementation, offering significant neuroprotective applications in brain pathologies.

## 1. Introduction

Among all the modifiable risk factors for age-induced cognitive impairment, diet composition plays a crucial role [1] due to its capability to induce structural and functional changes in brain connectivity [2], modulate the developmental programming of the brain and behavior [3], as well as affect cognition and emotion [4]. The well-known nutritional and culinary properties of mushrooms go hand in hand with their nature-based nutraceutical value, specifically known to mitigate age-related cognitive disturbances typical of neurodegenerative disorders, such as Alzheimer’s disease (AD).

The relationship between mushroom intake and mild cognitive impairment (MCI) has been investigated in cross-sectional and longitudinal studies carried out on well characterized cohorts at the Diet and Healthy Aging (DaHA) Research Centre, at the National University of Singapore (NUS). Interestingly, community-living elderly subjects who consumed more than two weekly portions of mushrooms (a portion being defined as 150 g) displayed up to 50 per cent reduced odds of developing MCI compared with aged participants who consumed mushrooms less than once per week. These research findings supports the likely role of mushrooms and their bioactive substances in promoting cognitive wellbeing, also delaying neurodegeneration [5,6].

Among all culinary medicinal mushrooms (MMs), *Hericium erinaceus* is a well-established candidate in promoting a “healthy brain” [7].

Despite the evidence obtained both in vitro and in preclinical studies describing the neuro-health properties of *H. erinaceus*, only few pilot clinical trials are available, although the scientific community feels the urgency and need to translate basic and preclinical research findings to clinical practices. This shift remains difficult for all MMs, possibly due to the different regulatory systems adopted by the Western and Eastern medicines regarding mushroom preparation. In China, a number of fungal glycan-based drugs were approved by the Chinese Food and Drug Administration (SFDA). Glycan-based drugs approved by the SFDA are extracted with hot water, either from cultured mycelium and/or sporophore, and have been used clinically in China since the 1980s [8].

Among them, glycans were extracted from the sporophores of *Lentinula edodes*, *Polyporus umbellatus*, *Tremella fuciformis*, and the cultured mycelium of *Trametes*
*versicolor*, *Poria cocos*, and *Grifola frondosa*. For example, Krestine (PSK) and PSP, two extracts from *T. versicolor*, the lentinan extracted from *L. edodes*, and the schizophyllan from *Schizophyllum commune* were recognized as drugs in the antitumor field since 1970 [9,10,11,12].

As far as we known, no drugs based on *H. erinaceus* components are still available for clinical use in China or Japan. In China, the traditional use of *H. erinaceus* is mainly based on its gastrointestinal properties, as a preventive or curative supplement in chronic gastrointestinal disease, such as Crohn’s disease. In Japan, the traditional use of *H. erinaceus* is more related to its known neuroprotective properties of the central nervous system [13].

MM extracts are defined as dietary supplements (DSs) in most Western countries following the directions of the World Health Organization (WHO) and the Dietary Supplement Health Education Act (DSHEA). Hence, clinical studies are not required for DSs to enter the market, and the consumer can take DSs without any medical prescription. DSs are consumed for their health benefits and their use has become part of complementary alternative medicine (CAM) or, even better, complementary integrative medicine (CIM). Paradoxically, this unregulated and large use of MMs as DSs has held back and delayed the development of proper clinical trials. Otherwise, in China and other Asian countries, the use of MMs, the so-called mycotherapy, has traditional and deep-seated heritages, and MM extracts are considered drugs. Concerning *H. erinaceus*, since ancient times, it has been widely employed as a neuroprotector in treating dementia in diverse Asian regions; nevertheless, clinical studies are still needed. 

Since 2014, Professor Wasser, one of the leading experts for MMs, hoped and urged the scientific community to bridge the gap between Western and Eastern medicine, since “it is to our advantage” [14,15].

Despite existing data revealing the nootropic effects of *H. erinaceus*, only a limited number of intervention studies have been conducted so far, while others are currently ongoing [16]. 

This review focuses on available and updated evidence disclosing the neuroprotective and nootropic effects of *H*. *erinaceus* dietary supplementation, trying to fill the gap between preclinical studies and clinical trials.

## 2. *Hericium erinaceus*

According to the main international databases, *Hericium erinaceus* (Bull.) Pers. belongs to the kingdom of fungi, Dikarya, Basidiomycota, Agaricomycetes, Russulales, and *Hericiaceae*, *Hericium* (http://www.indexfungorum.org accessed on 6 April 2023; http://www.mycobank.org accessed on 6 April 2023).

Taxonomy and phylogeny of the genus *Hericium* Pers. have been intensely debated in the last decades regarding its collocation within Agaricomycetes and the intra-genus discrimination of a single species. Thus, the *Hericium* genus is currently located in Russulales on the basis of both synapomorphic characters and molecular data [17]. Phylogenetically, the *Hericium* species are classified in the *Hericiaceae* family within the russuloid linage, very closely to *Laxitextum*, with steroid sporophores [18]. *Hericium* sporophores are clavariod or pileate, fleshy, and with a typical hydnoid hymenophore (this explains the genus name, which means hedgehog in Latin). The *Hericium* species have been classified taking into account different aspects, such as macro- and micro-morphological characters, host trees, geographical areas, and molecular data [19]. For species delimitation, phylogenetic analysis was carried out by comparing the rDNA internal transcribed tracer, ITS1 and ITS2 sequences. The comparison of ITS regions in nuclear ribosomal DNA is generally assumed to be the most appropriate approach for the molecular discrimination of fungal species, although it is regarded as not completely satisfactory, since a phylogenetic inference and molecular clades cannot be fully resolved by employing the ITS sequence only [20]. The 28S nuclear ribosomal large subunit rRNA gene (LSU) region (GeneBank), that has contiguity with the ITS region, is also used for identification, even though less frequently [21].

The monophyletic group *Hericium* was separated in two subtrees, showing that one includes *H. erinaceus*, *H. americanum*, *H. alpestre*, and *H. abietis* with a strong bootstrap support, whereas a separate tree resulted for the species *H. coralloides*, which is therefore phylogenetically more distant from *H. erinaceus* [21,22].

Most *Hericium* species have been reported from Europe, North and South America, and Asia, but new species are being described from different geographical areas [23,24,25,26].

*H. erinaceus* is also known by different common names, such as lion’s mane, monkey head mushroom, bearded tooth, bearded hedgehog or Yamabushitake. This species is mainly distributed in Asia, Europe, and North America, but new finds in different geographical areas have been recently reported [27]. In Europe, its spreading has reached Poland, Hungary, Belgium, Czech Republic, Netherlands, Slovakia, Romania, Bulgaria, Portugal, Turkey, Italy, Sweden, Denmark, Spain, and Ukraine [28]. In Italy, it is widely distributed and known in the Mediterranean area, which mainly includes central and southern Italy, besides the Sardinian and Sicilian islands [28]. 

*H. erinaceus* is widespread and abundant in a few areas, but very rare in others. The loss of habitat is the main threat for this species that is decreasing, above all, where it is rare. For this reason, it is included on the Red Lists of a few countries, and according to Kałucka and Olariaga (2019), it is classified as LC (least concern) [29].

Concerning its habitat, *H. erinaceus* preferentially grows on living, dying or dead broadleaved trees, mainly oak, walnut, beech, and others. It is generally found in knotholes and wounds. Figure 1 shows a sporophore of *H. erinaceus* found in Bedonia (province of Parma, Emilia Romagna, Italy) in October 2022, on a dying *Quercus cerris* (personal communication by Lorenzo Goppa).

*H. erinaceus* is morphologically characterized by a pileate, globose, hedgehog-like sporophore, with non-forking aculei 1–4 cm long [30]. The sporophore is more or less hirsute to tomentose, characterized by a hymenophore hydnoid, as well as an obtrusive odor. The external portion is whitish and fibrous. In particular, the color is white in the early stages of growth, then becomes cream, and at the end turns yellow–brown or brown when aged. *H. erinaceus* has a monomitic hyphal system: hyphae have clamps, thick and amyloid walls, and they are frequently ampullate and variable in size [28].

## 3. Untargeted Metabolomics as a New Approach for *Hericium erinaceus* Metabolite Detection

Untargeted metabolomics comprises methods and techniques used to analyze all small-molecules metabolites present in a biological sample in a particular physiological state, and it is currently one of the most rapidly evolving research fields. Metabolomics studies are characterized by high resolution, good quality and precision, both at qualitative and quantitative levels. The strength of this new approach is the possibility of investigating the entire set of metabolites of a biological sample. This new approach is very different from the traditional chemical characterization techniques, as it is typically targeted, therefore able to characterize only a single class of compounds or a single metabolite, defined a priori. With untargeted metabolomics, it is possible to take a “snapshot” of the sample, identifying and quantifying all the metabolites with a weight lower than 1000 Da [31]. The classes of investigated metabolites can include amino acids, flavonoids, nucleotides, organic acids, phenolic acids, phenylpropanoids, steroids, terpenoids, and unsaturated fatty acids [32]. This entire set of metabolites, defined as the metabolome, is extremely sensitive to the external environmental variations and/or growth conditions, so many recent works have aimed to clarify how the different sample preparation parameters can influence, and therefore improve, the metabolomic profile enhancement, for example, the synthesis of biocompounds of interest. Among the -omics approaches, which are genomics, transcriptomics, proteomics, and metabolomics, the last provides the closest representation of the metabolic phenotype.

Nowadays, several investigations apply untargeted metabolomics to vegetal samples [33], but only few of them performed it with mushrooms [34]. The analysis of the metabolome, i.e., the complete set of metabolites in the sample, could have multiple applications, such as the comparison of the metabolome profile among different strains, and/or under different growth condition, and/or at different developmental stages [35]. 

Therefore, metabolomics studies may be useful to investigate the presence and abundance of nootropic molecules in mushroom samples, but could also be useful to define the better experimental condition to boost them. 

Furthermore, the implementation of metabolomics investigations with those of chemotaxonomy may allow for the dissection of metabolite/metabolites patterns, which can help clarify the phylogenetic relationships of a genus with a multidisciplinary approach [35].

Species of the genus *Hericium* are known to produce a lot of secondary metabolites, many of which are responsible for the beneficial properties for humans and animals. 

In particular, *H. erinaceus* is well-known for its content in neuroprotective metabolites such as the terpenoids hericenones and erinacines [36], antioxidant and immunomodulating metabolites such as polysaccharides [37], and anti-aging metabolites such as the “longevity vitamin” L-ergothioneine (ERGO) [38].

Data about *H. erinaceus* compounds are continuously published, many of which were obtained by comparing with standard molecules, but research remains to be carried out. Untargeted metabolomics can be useful for comparing the bioactivity of different *Hericium* species, different *H. erinaceus* strains, and strains grown on different substrates or under different conditions [32].

## 4. *Hericium erinaceus* Bioactive Metabolites

Many mushrooms contain bioactive compounds in sporophores, cultured mycelium, cultured broth, and the primordium [39,40,41,42]. 

*H*. *erinaceus* mycelia and sporophore contain an exceptionally large amount of structurally different and potentially bioactive molecules, including about 70 different secondary metabolites [43]. These bioactive compounds can be subdivided in major classes of organic substances: polysaccharides, cyathan-type diterpenoids, and geranyl-resorcilate derivatives, alkaloids, lactones, and steroids (to depict the chemical structures, see [43,44]). Among these, erinacines, cyathin diterpenoids, and hericenones (C-H), benzyl alcohol derivatives extracted from the mycelium and sporophore have attracted scientists for their nootropic effects [45,46]. 

With the aim to provide a further step toward the standardization of procedures required in the accurate development process of a dietary supplements, the mycelium, primordium, and sporophore (wild type and cultivated) of an *H. erinaceus* strain (He2 MicUNIPV) from Italy were analyzed using a HPLC-UV-ESI/MS in our lab [42]. Erinacine A in the mycelium, and hericenones C and D in the sporophore were quantified through comparison with their standard molecules. For the first time, *H*. *erinaceus* primordium was also investigated to assess the presence of these molecules. Hericenes, structurally similar to hericenones at the molecular level, were detected in all analyzed samples. Compared to the wild type, a higher content of hericenones C and D was measured in cultivated sporophore. The comparison of these results, with those obtained studying another Italian *H. erinaceus* strain (He1 MicUNIPV), was then described. These findings led us to punctually select particular *H*. *erinaceus* strains being more suitable for mycelium production or sporophore cultivation, aimed at obtaining extracts which contain an elevated content of bioactive compounds (see Table 1). 

The content of erinacine A, measured in the two *H*. *erinaceus* Italian strains maintained at the MicUNIPV (Fungal Research Culture Collection, DSTA, University of Pavia, Italy), is between 105 and 150 µg/g, respectively, in lyophilized mycelium [42,47]. This value is comparable to that reported by Kryczkowski in improved submerged cultivation [48]. The differences in the content of hericenones C and hericenones D in the two dried sporophores, and erinacine A in the mycelium assessed in the two Italian *Hericium* species were also described (see Table 1). It has to be mentioned that gauged values were comparable with those reported in a previous work [49].

Ergothioneine (ERGO) is another “essential” nutrient, a diet-derived thiolated derivative of histidine, avidly taken up by some tissues owing to a specific high-affinity transporter, namely OCNT1, also called the ERGO transporter (ETT), or based on the encoding gene name, SLC22A4 [40,50,51,52]. The de novo synthesis of ERGO has been depicted in various fungi, including Basidiomycota [51,52]. ERGO is a powerful antioxidant in vitro and in vivo, acting as free radical scavenger and cytoprotective able to attenuate oxidative stress and nitrosamine damage induced by neurotoxic peptide, and reducing the beta-amyloid-induced apoptotic death in PC12 cells [53]. The intracellular antioxidant pathway involving the p38 MAPK cascade is activated by ERGO in rescuing cells to escape stress-induced apoptosis [53]. ERGO was demonstrated to mitigate cisplatin-induced nephrotoxicity targeting the apoptotic phenomena through p53 [54].

Dietary ERGO is efficiently and rapidly absorbed by OCTN1 from the small intestine and distributed to many body tissues, including the mouse brain [55], and it is highly retained after oral administration in humans [56]. In particular, increased ERGO, mediated by OCTN1 overexpression, was shown to have a cytoprotective effect in injured tissues (such as liver, heart, joints, and intestine), characterized by high oxidative stress and inflammation [57,58]. Therefore, the accumulation of the antioxidant ERGO in vivo could be an adaptive useful mechanism put into effect to minimize oxidative damage through an exogenous natural compound, also regulating its uptake and concentration. 

A bulk of literature data has also demonstrated the in vivo beneficial effects of ERGO on cognition and memory. Some papers revealed that ERGO treatment was able to prevent the cognitive deficits in murine models of AD, reducing amyloid plaques, oxidative stress, and rescued glucose metabolism [59,60]. Furthermore, other in vivo investigations showed that 88-day-lasting ERGO supplementation protected against memory and learning deficits in a model of accelerated senescence, reducing the oxidative stress. It was also reported that the combined treatment of ERGO and melatonin displayed higher beneficial effects compared to each single treatment [61]. Moreover, ERGO treatment reverted the learning and memory deficits induced by cisplatin in mice, probably through the inhibition of oxidative stress and lipid peroxidation in brain [62]. Additionally, it was demonstrated that the two-week-lasting ERGO treatment was sufficient to improve the response to behavioral tests in mice, increasing the expression of synapse formation markers in hippocampal neurons [63].

Concerning our investigations, the presence of ERGO was recently assessed in Italian *H*. *erinaceus* strains using HPLC-UV-ESI/MS. ERGO was measured in comparable amounts both in the mycelium and sporophore, with the *H*. *erinaceus* primordium showing the highest concentration (Table 1 and Table 2). 

Earlier studies demonstrated that a *H*. *erinaceus* blend, composed of mycelium and sporophore, partially rescued cognitive and locomotor frailty in a mouse model of physiological aging [39,42,47,64]. Recently, we studied the preventive effect during physiological aging on cognitive [40] and locomotor frailty [41] of the *H*. *erinaceus* primordium characterized by a high content of ERGO and the lack of erinacines and hericenones [42]. Notably, the ERGO amount was higher in *H*. *erinaceus* primordium compared to the mycelium and sporophore, as previously described. A similar quantity of ERGO was described by other groups in different *H*. *erinaceus* samples (Table 2). 

**Table 2 jof-09-00551-t002:** ERGO content in different *H*. *erinaceus* dried samples (from published papers).

Growth Stage	ERGO Content (mg/g)	Reference
Sporophore	0.96	[65]
Sporophore	1.12	[66]
Sporophore	1.6–3.7	[67]
Sporophore	1.31	[38]
Mycelium	0.38	[68]

## 5. *H. erinaceus In Vitro* Preclinical Studies and Molecular Mechanism Underlying the Nootropic Effects

In the last decade, an increasing body of evidence has shown that *H*. *erinaceus* administration can improve neuronal conditions in animal models of neurodegenerative diseases such as AD, depression, Parkinson’s disease, and spinal cord injury. Indeed, preclinical trials have successfully demonstrated that *H*. *erinaceus* bioactive compounds significantly improve cognitive function and rescue behavioral deficits [16]. Several in vitro and in vivo studies provided evidence demonstrating the nootropic and neuroprotective effects displayed by erinacines. In particular, in vivo investigation in mouse models revealed that erinacine A in the mycelia confers nootropic effects and reduces oxidative stress against stroke [46], AD [69], depression [70], and aging [71]. On the other hand, the neuroprotective role played in vivo by hericenones from the *H*. *erinaceus* sporophores is still controversial, even though oral sporophore supplementation exerted a nootropic action in pilot clinical trials, as reported in the Section 8 [72,73,74]. Nonetheless, few studies have been conducted thus far on the bioavailability and tissue distribution of these relatively hydrophobic metabolites with a low molecular weight. Erinacine A and S distribution in the CNS has been analyzed after oral administration in rats, revealing the presence of the bioactive compound in the brain already one hour after ingestion, suggesting a passive diffusion of erinacines A and S through the blood-brain barrier as the dominant transport method [75,76]. Concerning the mechanisms of action involved in the nootropic effect, erinacine A from *H*. *erinaceus* mycelium stimulates the nerve growth factor (NGF) synthesis and promotes the NGF-induced neurite outgrowth stimulation, but also protects neuronally differentiated PC12 pheochromocytoma cells against NGF deprivation [77]. In vivo, erinacine A successfully upregulated the NGF level in the hippocampus and locus coeruleus of rats [78]. Furthermore, erinacine A displayed a protective role in vitro, preventing glutamate-insulted apoptosis in PC12 cells [79]. Moreover, an in vivo study conducted using a rat model of global ischemic stroke further revealed the *H*. *erinaceus* defensive function, showing the inhibition of inducible NO synthase (iNOS), phosphorylation of p38 MAPK and CHOP, and the reduction of acute inflammatory cytokine levels [46]. Lately, the beneficial regenerative effect of *H*. *erinaceus* has been described in a mouse peripheral nerve injury model, demonstrating that, compared to NGF, *H. erinaceus* possesses higher neuroprotective and neurotogenic action, improving axonal regeneration ability [80]. Additionally, an aqueous extract of *H. erinaceus* was demonstrated to stimulate neurite outgrowth in NG108-15, a neuroblastoma–glioma cell line, with a synergistic interaction with exogenous NGF [81]. 

In 1321N1 human astrocytoma cells, hericenones C, D and E from the *H. erinaceus* sporophore, failed to promote NGF gene expression [82], while only isolated hericenone E, among all the present hericenones, was able to stimulate NGF-mediated neurite outgrowth via the MEK/ERK and PI3K-Akt signaling pathways in PC12 cells [83]. The involvement of the MEK/ERK intracellular signaling pathway has also been described for isolated erinacine A [84].

Within the elderly population (over 65 years old), the functional deficiency of NGF is related to progressive neurodegeneration and dementia-like diseases. Natural compounds able to induce the NGF biosynthesis are considered potentially effective against cognitive disturbances, e.g., dementia [45]. In this context, the neuroprotective role of *H. erinaceus* extracts is crucial and typically embraces five major aspects, one related to each other: aging, memory, dementia, depression, and AD [39,40,41,47,73,85,86,87].

During the last years, we focused our research on this key topic, firstly addressing whether the *H. erinaceus* nootropic effect could be discernible in wild-type animals also, and during their physiological aging [39,40,41,47,64,88]. 

In hippocampal brain slices obtained from wild-type middle-aged mice supplemented for two months with a *H. erinaceus* blend made of mycelium and sporophore ethanol extracts, an increase in glutamatergic neurotransmission was recorded in the synapses between mossy fiber and granule cells, both in spontaneous and evoked post-synaptic currents [88]. The increased efficiency of neurotransmission fitted with the increase in recognition memory, a declarative explicit form of long-term memory fundamental for human personality and behavior. Interestingly, another *H. erinaceus* blend made of mycelium and sporophore, containing defined amount of erinacine A, hericenones C and D, was able to partially revert the cognitive and locomotor frailty index during physiological aging [47]. Additionally, an increase in proliferating cell nuclear antigen (PCNA) and doublecortin (DCX) measured in the hippocampus and cerebellum of *H. erinaceus*-supplemented mice supported the occurrence of neurogenesis in elderly frail mice [47]. Accordingly, one-month-long administration of *H. erinaceus* extracts in adult wild-type mice significantly increased the expression of PCNA and Ki67 in hippocampal progenitor cells, suggesting an increase in their proliferation, and hence, an increase in neurogenesis [89].

In the cerebellar cortex, lobules VI–VIII are particularly sensitive to aging-induced locomotor and cognitive decline [90]. Our recent in vivo study on the cerebellum demonstrated that a two-month oral supplementation was able to ameliorate age-induced cerebellar alterations (e.g., volume reduction, molecular layer thickness decrease, and shrunken neurons), also decreasing inflammation, oxidative stress, and reactive gliosis. These findings supported the neuroprotective action played by *H. erinaceus*, which parallelly increased a key longevity regulator [39].

Further, we studied the preventive effect of *H. erinaceus* primordium (He2 strain) extract containing a high amount of ERGO, on cognitive and locomotor decline during physiological aging in wild-type animals. Eight-month dietary supplementation with the He2 primordium extract (starting at the adulthood phase of the mouse lifespan and lasting until senescence) was able to reduce both the locomotor decline, as well as oxidative stress in the cerebellum. Therefore, we demonstrated that ERGO-rich He2 primordium exerted a neuroprotective and preventive action, ameliorating/mitigating/reverting age-dependent impairments [41].

Additionally, the same extract was demonstrated to decrease oxidative stress and inflammation in the hippocampus, also preventing recognition memory decline and increasing the expression of specific receptors crucially involved in glutamatergic neurotransmission [40].

Concerning the evaluation of the neuroprotective effects in preclinical studies using animal models, erinacine A-enriched *H*. *erinaceus* mycelium (EAHEM) was tested in AD APP/PS1 mice. A recovery in cognitive disability was described in diverse behavioral tests, i.e., passive avoidance and active shuttle avoidance tasks. Additionally, a *H. erinaceus*-induced lowering of oxidative stress and inflammation levels was reported, paralleled by the decrease in amyloid plaque aggregation [36].

Moreover, an investigation in transgenic APP/PS1 mice revealed that a 30-day short-term EAHEM feeding induced (i) a decrease in Aβ plaque burden, and (ii) the prevention of recruitment and activation of plaque-associated astrocytes and microglia. Additionally, the increased NGF/proNGF ratio was paralleled by (i) an enhanced proliferation of neurons progenitors and (ii) an increased neuronal proliferation in the dentate gyrus [87]. Several studies proved that the *H. erinaceus* mycelium ameliorates AD-related pathologies. In particular, isolated erinacine A and S displayed beneficial effects in the cerebrum of APPswe/PS1dE9 transgenic mice. In fact, a 30-day-long administration of erinacine A and S attenuated cerebral plaque loading by inhibiting plaque growth, diminishing glial cell activation and promoting hippocampal neurogenesis [69].

In another AD animal model induced by an aluminum (AlCl_3_) intraperitoneal injection, *H. erinaceus* administration reduced neuronal degeneration in the rat hippocampus, also lessening oxidative and inflammatory alterations. Additionally, at a molecular level, *H. erinaceus* reduced the β-amyloid accumulation, aberrant APP overexpression, phosphorylated Tau, and the activation of the NLRP3 inflammasome components. Finally, *H. erinaceus* had protective effects on behavioral changes, increasing the discrimination ratio in novel object recognition tasks and animal permanence in target quadrants [91].

The effects of *H. erinaceus* on amyloid β(25-35)-peptide, intracerebroventricularly injected, were assessed in peptide-induced learning and memory deficits in mice. *H. erinaceus* prevented impairments of spatial and visual recognition memory induced by amyloid β(25-35)-peptide, tested by the Y-maze novel-object recognition behavioral tests [92].

The neuroprotective effects of the *H. erinaceus* mycelium polysaccharide-enriched aqueous extract was tested in an AD mouse model (AlCl₃ combined with d-galactose-induced). *H. erinaceus* aqueous extract administration ameliorated the endurance time in the rotarod test, enhanced the horizontal and vertical movements in the activity test, and decreased the escape latency time in the water maze test. The mechanism involved was an *H. erinaceus*-induced improvement in the central cholinergic system function, also responsible for the dose-dependent enhancement of acetylcholine (Ach) and choline acetyltransferase (ChAT) concentrations, both in the serum and hypothalamus [93]. 

In the SAMP8 (senescence accelerated mouse prone 8) model of accelerated senescence and APP/PS1 model of AD, *H. erinaceus* ameliorated learning and memory abilities. These behavioral changes were paralleled by a significant reduction in brain tissue swelling, neuronal apoptosis, and the down-regulation of Tau and Aβ1-42 [94].

The mechanism by which EAHEM delays the brain cognitive decline during aging was assessed using the SAMP8 mouse model. iNOS, Thiobarbituric acid reactive substances (TBARS), and 8-Hydroxy-2′-deoxyguanosine (8-OHdG) brain levels significantly decreased in mice supplemented with EAHEM, with a dose-dependent recovery of cognitive skills, such as learning and memory. Moreover, in an animal model of ischemic stroke, EAHEM reduced the ratio of cerebral infarction [46]. 

Table 3 summarizes the cited effects of the *H. erinaceus* metabolites in terms of targets and pathways, and Figure 2 resumes the principal effects of bioactive molecules (ERGO, erinacine A, and hericenone E) contained within *H. erinaceus*.

## 6. *H. Erinaceus* and Its Potential within the Gut-Microbiome–Brain Axis

The gut microbiota and the brain communicate with each other through several mechanisms, and this bidirectional communication is named the gut-microbiome–brain axis. Today, even if the processes leading to the communication between the gut microbiota and the brain are not yet clear, it is known that this communication can be direct, through the intestinal nervous system and the vagal nerve, or indirect through the stimulation of the release of several molecules such as short-chain fatty acids, amino acids, vitamins, hormones, and neurotransmitters that affect metabolism and the immune system, which in turn affects the integrity of the blood-brain barrier and brain function. On the other hand, it was demonstrated that variations in the central nervous system neurotransmitter concentrations influence the proliferative activity of several gut bacteria [95,96,97,98].

The gut-microbiome–brain axis lies at the intersection between microbiology and neuroscience, looking for the correlation between the ecological fitness of our microbial communities and neurological/neurodegenerative diseases, e.g., schizophrenia, autism spectrum disorder, depression, anxiety, AD, and multiple sclerosis [99]. The gut-microbiome–brain research gives us new and intriguing challenges/opportunities to implement currently existing therapeutic approaches for the treatment of brain diseases. The cause–effect relationship and the bidirectional interaction between the gut and complex human diseases related to brain development, mood, and neurodegeneration is still an open and fascinating debate in science [100]. Several recent papers demonstrated that treatment with prebiotics or probiotics could alter neuroplasticity and behavior, influencing the gut microbiota composition [98,101,102,103,104,105,106]. Beta-glucan polysaccharides, as fibers, could elicit a direct prebiotic effect and/or an immune-modulating effect. Both mechanisms could be involved in changing the gut microbiome composition [107,108]. *H. erinaceus* is known to ameliorate gastrointestinal diseases, and a widespread interest concerns the possible effects of *H. erinaceus* polysaccharides (HEPs) in modulating the gut microbiota (GM) (for a review see [109]), and Table 4 summarizes the ones discussed in this review. 

The digestibility and fermentation of HEPs and their influence on the GM composition was investigated in vitro on human fecal microbiota fermentation. The SCFA content, as well as the relative abundance of SCFA-producing bacteria (for detail see Table 4), were significantly enhanced. Furthermore, HEPs reduced several opportunistic pathogenic bacteria (Table 4) [110].

The role of two *H. erinaceus*-extracted polysaccharides in maintaining the integrity of the intestinal barrier was investigated both in vitro and in vivo (using murine models). One of the two polysaccharides significantly increased TEER and paracellular permeability. Both polysaccharides were able to significantly (i) increase the expression of occludin, ZO-1, ZO-2, claudin-3, claudin-4, and MUC2, (ii) decrease claudin-2, and (iii) alter the composition of the gut microbiota, boosting *Bacteriodetes*, *Firmicutes* and lessening *Klebsiella* and *Shigella* [111]. 

In a pilot clinical study on 13 healthy adults, seven days of *H. erinaceus* dry powder in submerged culture displayed beneficial health effects on the GM detected as 16S ribosomal RNA. In particular, *H. erinaceus* upregulated the relative abundance of some short-chain fatty acid (SCFA)-producing bacteria and downregulated some pathobionts (Table 4) [112].

Prebiotics may restore the gut microbial imbalance during the aging process, as evidenced by three different studies. Briefly, (i) by using an in vitro batch culture fermentations and fecal inocula from elderly donors, *H. erinaceus* increased the production of SCFA [113]; (ii) in aged dogs, the GM community improved immunity and anti-obesity [114]; (iii) in middle- and old-aged mice, the relative abundances of *Akkermansiaceae* and *Lachnospiraceae* significantly enhanced, while the relative abundance of *Bacteroidaceae* and *Rikenellaceae* declined [115].

Further, in cyclophosphamide (CTX)-immunodeficient mice, HEP improved the GM structure and inhibited the CTX-induced GM dysregulation [110].

In two different animal models of induced Ulcerative colitis (UC), *H. erinaceus* mycelium beneficial action was reported. EP-1, a purified unique polysaccharide, remarkably changed the GM, increasing SCFA production, also showing antioxidant/anti-inflammatory effect, and enhancing immune activities [116]. A novel low-weight polysaccharide altered the GM composition, and promoted functional shifts and structure by increasing *Akkermansia muciniphila*. Moreover, HEP10 treatment significantly suppressed the activation of the NLRP3 inflammasome, NF-κB, AKT and MAPK pathways [117]. 

Again, HEP was administrated for 2 weeks in rats suffering from induced inflammatory bowel disease (IBD). Tissue damage scoring in colonic mucosa was ameliorated, and the GM composition significantly changed compared to that determined in the untreated group [118]. A single-band protein (HEP3) isolated from *H. erinaceus* played a prebiotic role in the case of disproportionate use of antibiotics in IBD [119].

*H. erinaceus* (HEP) was administered in a xenografted mouse model of cancer with the chemotherapy drug 5-Fluorouracil (5-Fu), with the purpose of identifying new potential prebiotic bacteria for complementary and integrative antitumor treatment. HEP bettered the dysbiosis induced by 5-Fu, as it inhibited certain aerobic and micro aerobic bacteria, also increasing some probiotic bacteria (Table 4) [120]. 

Despite (i) all the reported *H. erinaceus* prebiotic effects on the GM, described during aging in diverse diseases, e.g., ulcerative colitis, IBD, and in integrative cancer therapy, and (ii) the *H. erinaceus* effects on the CNS, it has to be underlined that, until now, no data were available about the possible *H. erinaceus*-induced outcomes on the gut-microbiota–brain axis.

**Table 4 jof-09-00551-t004:** Summary of the effects of *H. erinaceus* treatment on the GM in different models.

Model/Condition/Disease	Treatment (Doses and Duration)	Increased Bacteria	Decreased Bacteria	Other Effects	Reference
In vitro (after simulated gastrointestinal digestion and 24-h-fermentation) on fermentation broth of human feces	HEPs ^1^ (HEP-30, HEP-50, and HEP-70)	SCFA ^2^-producing bacteria: *Lactobacillus*, *Faecalibacterium*, *Bifidobacterium*, *Blautia*, *Butyricicoccus*	Pathogenic bacteria: *Klebsiella*, *Escherichia-Shigella*, *Enterobacter*	-	[110]
In vitro (static batch culture 24 h fermentations and fecal inocula from elderly donors)	HEBS ^3^ and HEOLRP ^4^	-	-	Increasing propionate and butyrate levels.	[113]
Healthy adults (30.0 ± 4.9 years old)	1 g of *H. erinaceus* powder three times a day (7 days of washout)	SCFA-producing bacteria: *Eubacterium rectale*, *Faecalibacterium prausnitzii*, *Kineothrix alysoides*, *Gemmiger formicilis*, *Fusicatenibacter saccharivorans*	Pathobionts: *Bacteroides caccae*, *Streptococcus thermophilus*, *Romboutsia timonensis*	Increasing in GM ^5^ alpha diversity. GM changes related to blood ALP ^6^, LDL ^7^, UA ^8^, and CREA ^9^.	[112]
Aged dogs	*H. erinaceus* 0.8 g per bw ^10^ with daily diets for 16 weeks	*Bacteroidetes* (order *Bacteroidales*)	*Firmicutes: Streptococcus*, *Tyzzerella*, *Campylobacteraceae* (genus *Campylobacter*)	-	[114]
Middle (8-week-old) and old (6-month-old)-aged mice	HEPs from the *H. erinaceus* sporophore 0.5 or 1 g/bw/day by gavage for 28 days	*Lachnospiraceae*, *Akkermansiaceae*	*Rikenellaceae*, *Bacteroidaceae*	Anti-inflammatory activity and immunomodulatory activity.	[115]
Acetic-acid-induced UC ^11^ adult rats	EP-1 ^12^ 0.6 and 1.2 g/kg by gavage for 11 days	SCFA producers	-	Antioxidant, anti-inflammatory, and immunoregulatory activities. Restoring a normal GM.	[116]
DSS ^13^-induced UC C57BL/6 adult male mice	HEP10 ^14^ 50, 100, or 200 mg/kg by gavage for 7 days	*Akkermansia muciniphila*.	*Proteobacteria*	Restoring GM-alpha and -beta diversities. Anti-inflammatory activity in colons.	[117]
IBD ^15^ adult rats and mice	HE/HEP3 100 mg/(kg * day) by gavage for 14 days	Anti-inflammatory bacteria, *Corynebacterium*, *Bacteroides*, *Enterobacter*, *Acinetobacter*, *Desulfovibrio*, *Lactobacillus*	Pro-inflammatory bacteria	Anti-inflammatory activity. Improving the GM composition.	[118,119]
CTX ^16^-immunodeficient adult mice	HEP and FHEP 300 mg/kg/day by gavage for 21 days	*Bacteriodetes*, *Firmicutes*	*Klebsiella* and *Shigella*	Anti-inflammatory activity. Improving TJ ^17^ and MUC ^18^ expression.	[111]
CTX-immunodeficient adult mice	HEPs 75, 150, and 300 mg/kg by gavage for 4 weeks	SCFA-producing bacteria, *Alistipse*, *Muribaculaceae*, *Lachnospiraceae_NK4A136_group*, *Lachnospiracea*, *Ruminococcaceae*, *Ruminococcaceae_UCG-014*	*Lactobacillus*, *Bacteroides*, *Alloprevotella*	Improving the body weight and immune organ index. Increasing OTUs and adjusting GM-alpha and -beta diversities.	[121]
Xenograft adult male Balb/C mice, implanting with 100,000 CT-26 wt cancer cells	HEP 100 mg/kg/day per os for 21 days	Probiotic bacteria: *Bifidobacterium*, *Gemellales*, *Blautia*, *Sutterella*, *Anaerostipes*, *Roseburia*, *Lachnobacterium*, *Lactobacillus*, and *Desulfovibrio*	*Parabacteroides*, *Christensenellaceae*, *Anoxybacillus*, *Staphylococcus*, *Aggregatibacter*, *Comamonadaceae*, *Desulfovibrionaceae*, *Sporosarcina*, *Planococcaceae*, *Aerococcaceae*, *Flavobacteriaceae*, and *Bilophila*	Reducing the 5-Fu-induced GM dysbiosis, suppressing tumor growth, and inhibiting inflammatory markers.	[120]

Abbreviations: ^1^
*H. erinaceus* polysaccharides (HEPs), ^2^ shorty-chain fatty acid (SCFA), ^3^
*H. erinaceus* LGAM 4514 in 100% beech sawdust (HEBS), ^4^
*H. erinaceus* LGAM 4514 in olive pruning residues (HEOLRP), ^5^ gut microbiota (GM), ^6^ alkaline phosphatase (ALP), ^7^ low-density lipoprotein (LDL), ^8^ uric acid (UA), ^9^ creatinine (CREA), ^10^ body weight (Bw), ^11^ ulcerative colitis (UC), ^12^ purified unique polysaccharide isolated from *H. erinaceus* mycelium (EP-1), ^13^ dextran sulfate sodium (DSS), ^14^ low-weight polysaccharide from *H. erinaceus* with Mw: 9.9 kDa (HEP10), ^15^ intestinal bowel disease (IBD), ^16^ cyclophosphamide (CTX), ^17^ tight junctions (TJ), ^18^ mucin (MUC).

## 7. *H. Erinaceus* Pilot Clinical Trials on Antidepressants

One of the foremost causes of global disease burdens is depression, a severe neuropsychiatric disorder. Currently, a number of antidepressants are available, but their efficacy is only just adequate and the side effects are very common. Four pilot clinical studies investigating the potential antidepressant action played by *H. erinaceus* were conducted so far on a small number of patients (for a review, see [122]). 

Nagano and colleagues (2010) examined the clinical effects of *H. erinaceus* on menopause, depression, and sleep quality in 30 females (average age of 41.3 years) over a period of 4 weeks [72]. The consumption of cookies containing 0.5 g of sporophore powder alleviated the symptoms of depression, frustration, anxiety, and palpitation. However, this study was gender-specific by design, related to menopause, and a small population was used, making the conclusions only partially relevant [72].

In 2014, Inanaga and collaborators described a case-report of an 86-year-old male patients affected by recurrent depressive disorder which presented an improvement in neurocognition after treatment with a standardized extract of amycenone and hericenones, Amyloban^®^ 3399. However, mirtazapine, an antidepressant drug, was also administered together with Amyloban^®^ 3399, making it difficult to assess whether the alleviation of depression symptoms was a result of mirtazapine or Amyloban^®^ 3399, or both [123]. 

Additionally, a pilot study on eight female undergraduate students with sleep disorder demonstrated that a 4-week administration of Amyloban^®^ 3399 was associated with an improvement in anxiety and sleep quality, measured by the increase in the salivary level of free 3-methoxy-4-hydroxyphenylglycol, a biological index of anxiety disorders [124]. 

Lastly, epidemiological data indicated that obese subjects have an increased risk of developing mood disorders and vice versa, giving a bidirectional relationship between obesity and depression. In a first clinical study, patients affected by overweight or obesity were evaluated through self-assessment questionnaires, and then recruited only when positive for one or more administered tests, including Symptom Checklist-90, Zung’s Anxiety and Zung’s Depression Self-Assessment Scales, and the binge eating scale. This study was conducted at the Department of Preventive Medicine, Luigi Devoto Clinic of Work, Obesity Centre, at the IRCCS Foundation Policlinico Hospital of Milan (Italy). The 4-week oral supplementation using a *H. erinaceus* blend made of 80% mycelia and 20% sporophore, water, and ethanolic extract alleviated symptoms of depression, anxiety, and sleep disorders in patients [85]. This observation was linked to an increase in the serum level of proBDNF, the precursor form of BDNF, and in the proBDNF/BDNF ratio, without any significant change in the BDNF circulating level [85]. 

## 8. *H. Erinaceus* Pilot Clinical Trials on Cognitive Functions

Few clinical studies have been conducted on *H. erinaceus* nootropic effects [125].

A pilot interventional study was conducted on 31 subjects over 50 years old that were healthy, and with normal cognitive functions after *H. erinaceus* oral administration. The dry powder of *H. erinaceus* sporophore (0.8 g four times a day, for 12 weeks) was tested in this randomized, double-blind placebo-controlled parallel-group comparative study [74]. Three different tests were used to assess the effects of *H. erinaceus* on cognitive functions: Mini Mental State Examination (MMSE), standard verbal-paired-associate learning test, and Benton visual retention. The first is an assessment of associative and episodic memory. *H. erinaceus* oral intake significantly improved cognitive functions and prevented deterioration. The authors concluded that *H. erinaceus* oral consumption is safe and seems a convenient method for preventing dementia so far. In a college-age cohort (*n* = 24), the 4-week-long ingestion of 10 g *H. erinaceus*/day did not elicit any statistically significant changes of either cognition nor metabolic flexibility markers [86].

Mild cognitive impairment (MCI) is typically considered the initial stage between the cognitive decline of normal aging and the more serious decline of dementia [126].

The efficacy of *H. erinaceus* oral administration for improving cognitive impairment was tested in a double-blind placebo-controlled parallel-group clinical trial of thirty Japanese patients diagnosed with MCI [73]. MCI was tested with the cognitive function scale, based on the Revised Hasegawa Dementia Scale (HDS-R). *H. erinaceus* was administrated for 16 weeks as four 250 mg tablets containing 96% of sporophore dry powder, three times a day. After 4 weeks of intake, cognitive functions were monitored. Compared with the placebo group, the *H. erinaceus* group showed significantly increased scores of the cognitive function scale (score value 22–25 vs. 30); meanwhile, after intake termination, the scores decreased significantly. The safety of *H. erinaceus* intake was tested through laboratory analysis and confirmed by the absence of adverse outcomes. Nonetheless, the bioactive compounds contained within *H. erinaceus* tablets used for this study have not been extensively addressed [73].

Patients with mild AD (*n* = 49) were investigated for their cognitive functions in a double-blind placebo-controlled parallel-group study. The oral administration consisted of 15 mg erinacine A per day as dried erinacine-A-enriched *H. erinaceus* mycelia (EAHE) [127]. Cognitive assessments, ophthalmologic examinations, biomarker collection, neuroimaging, and laboratory test analyses to check for safety were followed throughout. In the EAHE group, after 49 weeks of EAHE intervention, a significant improvement in the MMSE score was observed, and a significant Instrumental Activities of Daily Living score difference was found between the two groups [127].

In relation to blood markers, only the placebo group displayed a significantly lower amount of calcium, albumin, apolipoprotein E4, hemoglobin, and BDNF, and parallelly showed a significant increase in alpha1-antichymotrypsin and amyloid-beta peptide 1–40. Furthermore, in neuroimaging of the dominant hemisphere, the mean apparent diffusion coefficient values obtained from the arcuate fasciculus region was significantly enhanced in the placebo group. Any significant difference was determined in the EAHE group after intervention compared to the baseline level. Moreover, ADC values from the parahippocampal cingulum region were significantly diminished in the EAHE group [127].

Figure 3 shows the timeline of the principal published papers related to the effects of *H. erinaceus* on patients with mood disorders or some cognitive impairments.

## 9. Conclusions

To sum up, the mass of scientific data reported leaves no doubt about the nootropic properties of *H. erinaceus*. It also clearly appears that different nootropic substances are present in both the *H. erinaceus* mycelium and sporophore. The efforts of our laboratory aimed to obtain standardized extracts by performing chemical analysis and/or through a metabolomics approach on specific bioactive compounds. We suggest that this could be a general practice to follow. In our in-depth thinking, among the bioactive compounds, ERGO is one of the most promising. This research activity is needed to obtain dietary supplements, functional foods, or drugs to promote a healthy brain. Furthermore, the pharmacologically active ingredients, as well as the mode of action, require further in-depth clinical investigation.

Erinacine A contained within the mycelium is one of the key components responsible for the *H. erinaceus* nootropic effects, as confirmed both in vitro, as well as in preclinical investigations in vivo. Moreover, sporophore extracts were proven to possess nootropic or neuroprotective power, too. In the above-reported pilot clinical studies, sporophore oral consumption was checked as a beneficial food, revealing an improvement of specific cognitive functions. To increase the array of nootropic substances even more, the free radical scavenger and cytoprotective powerful antioxidant ERGO, contained in both the mycelium and sporophore, attenuates oxidative stress and nitrosamine damage in an injured or aging brain. 

More intriguingly, we want to underline that following *H. erinaceus* intervention, on top of the improved pathological phenotype consistently evidenced in clinical trials, the downstream molecular pathway responsible for the phenotypic rescue involves, as evidenced in vitro and in preclinical studies, the NGF and BDNF. However, we highlight the urgent need for additional in-depth clinical studies to clarify the mode of action by which *H. erinaceus* compounds are able to promote functional neurophysiological recovery in human, and assess the link between *H. erinaceus*, microbiota, and cognition in more detail.

## Figures and Tables

**Figure 1 jof-09-00551-f001:**
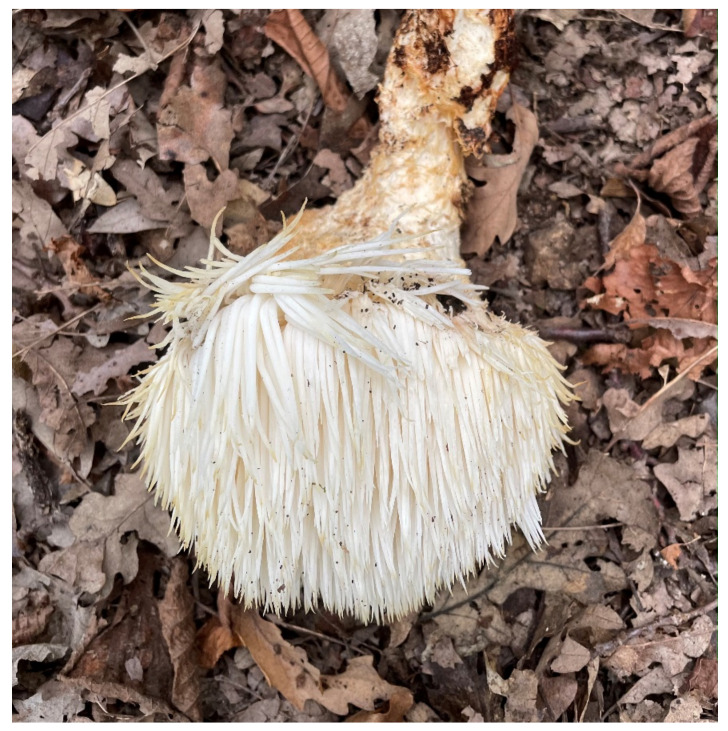
*H. erinaceus* grown on *Q. cerris*: the collected sporophore lies on the ground (photo by L. Goppa).

**Figure 2 jof-09-00551-f002:**
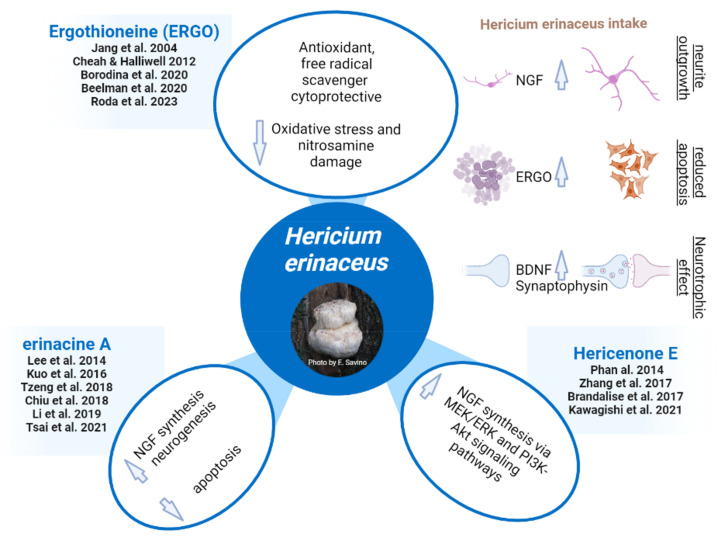
*H. erinaceus* potentials (photo by E. Savino) [40,45,46,50,51,52,53,69,70,71,77,83,88].

**Figure 3 jof-09-00551-f003:**
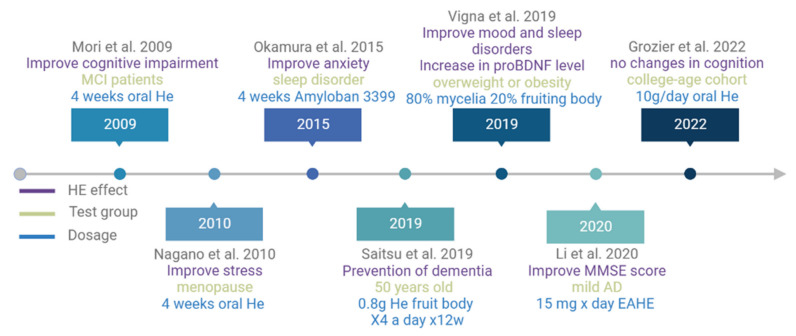
Timeline of the published papers relating the effects of *H. erinaceus* in clinical studies [72,73,74,85,86,124,127].

**Table 1 jof-09-00551-t001:** Erinacine A, hericenone C, hericenone D, ergothioneine in the sporophore, mycelium, and primordium of Italian strains, He1 and He2 (data from [39,40,41,42,47]).

	He1		He2		
	Sporophore	Mycelium	Sporophore	Mycelium	Primordium
Erinacine A (µg/g)	-	150	-	105	-
Hericenone C (µg/g)	500	-	1560	-	-
Hericenone D (µg/g)	<20	-	188	-	-
L-Ergothioneine (µg/g)	340	580	2400 (unpublished data)	940 (unpublished data)	1300

**Table 3 jof-09-00551-t003:** Summary of the main in vitro and preclinical studies on *H. erinaceus* in the CNS. In the table the up arrows indicate an increase, and the down arrows indicate a decrease.

*H. erinaceus* Bioactive Compound	Target	Pathway/Intracellular Mechanism	Effect	Reference
**Erinacine A**/ **erinacine A-enriched *H*. *erinaceus* mycelium (EAHEM)**	NGF ^1^, ↑ NGF/proNGF ratio ↓ iNOS ^2^ ↑ IDE ^5^ ↓ 8-OHdG ^7^, iNOS, and TBARS ^8^ levels in mice brain	TrkA/Erk1/2 pathway Others? Phosphorylation of p38 MAPK ^3^ and CHOP ^4^. ? ?	Neuroprotection and nootropic in vitro. ↑ Neurite outgrowth. ↑ Neuronal proliferation. ↓ Cognitive decline during aging in a SAMP8 mouse model. Protection against stroke in rats. ↓ Inflammation. ↓ Cerebral β-amyloid plaque loading in AD ^6^ APPswe/PS1dE9 transgenic mice. ↑ Cognitive performances in AD APP/PS1 mice. ↓ Oxidative stress, inflammation, brain Aβ plaque number.	[46,77,78,79,84] [46] [69] [36]
**Erinacine S**	↑ IDE	?	↓ Cerebral β-amyloid plaque loading in AD APPswe/PS1dE9 transgenic mice.	[69]
**Hericenone E**	↑ NGF	MEK/ERK and PI3K-Akt pathways	↑ Neurite outgrowth in vitro.	[83]
**Mix of Erinacine A and Hericenone C and D (sporophore and mycelium *H. erinaceus* extract)**	↑ Hippocampal and cerebellar PCNA ^9^ and DCX ^10^ expression ↓ Cerebellar expression of IL-6 ^11^, iNOS, COX-2 ^12^, and SOD-1 ^13^	? ?	↑ Neurogenesis and cognitive performances in aged frail mice. Inflammation and ↓ Oxidative stress in the cerebellum. ↑ Locomotor functions in aged frail mice.	[47] [39]
**Ergothioneine-enriched primordium**	↓ Cerebellar expression of iNOS and COX-2 ↓ Hippocampal expression of IL-6, Nrf2 ^14^, SOD-1, COX-2, and iNOS. ↓ Hippocampal NMDAR1 ^15^ and mGlutR2 ^16^ receptors expression	? TGF-beta1 ^17^	↓ Oxidative stress in the cerebellum and hippocampus, and of locomotor aging-related decline in frail mice. ↓ Hippocampal inflammation and oxidative stress. ↑ Glutamatergic neurotransmission. ↓ Cognitive aging-related decline in frail mice.	[41] [40]

Abbreviations: ^1^ nerve growth factor (NGF), ^2^ inducible nitric oxide synthase (iNOS), ^3^ mitogen-activated protein kinase (MAPK), ^4^ homologous protein (CHOP), ^5^ insulin-degrading enzyme (IDE), ^6^ Alzheimer’s disease (AD), ^7^ 8-hydroxy-2’-deoxyguanosine (8-OHdG), ^8^ thiobarbituric acid reactive (TBARS), ^9^ proliferating cell nuclear antigen (PCNA), ^10^ doublecortin (DCX), ^11^ interleukin 6 (IL-6), ^12^ cyclooxygenase-2 (COX-2), ^13^ superoxide dismutase 1 (SOD-1), ^14^ nuclear factor erythroid 2–related factor 2 (Nrf2), ^15^ N-methyl-D-aspartate receptor 1 (NMDAR1), ^16^ metabotropic glutamate receptor 2 (mGlutR2), ^17^ transforming growth factor beta 1 (TGF-beta1).

## Data Availability

Not applicable.

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
