# Peer review of "Hericium erinaceus in Neurodegenerative Diseases: From Bench to Bedside and Beyond, How Far from the Shoreline?"

_jof, 2023, doi:10.3390/jof9050551_

Round 1

Reviewer 1 Report

The paper submitted for review is extremely interesting and presents a very important guess, which is the effect of Hericium erinaceus species on brain function. It is known that the authors have made a great effort to present relevant knowledge of clinical research related to this species. I recommend publishing the paper after making the changes presented below:

Line 77 - He - why isn't there an expanded abbreviation in parentheses, or an explanation of the abbreviation? Please correct.

Line 78 - No citations, please complete.

Line 89-90 - It should add: species and family : „Phylogenetically, Hericium species is classified in the Hericiaceae family within the russuloid linage, very closely to Laxitextum, with steroid sporophore [11].” This should be corrected throughout the manuscript.

Line 117 - There should be information on whether it is a protected species in danger of extinction. Please add.

Line 120 - He is an abbreviation, not a Latin name. Remove the italic next to this abbreviation throughout the manuscript.

Line 121 - No citations, please complete.

Line 168 - Citation [34,35] should be at the end of the sentence. Check and correct throughout the manuscript.

Line 261, 290, 293, 386 - No citations, please complete. Each completed paragraph should end with a citation, authors must complete this in many places in the manuscript!!!

Line 263 – Citations should be at the end of the sentence!

Table 4 - The list of abbreviations should be under the table to make it more readable.

Line 495 - No citations, please complete.

Line 507 - No citations, please complete.

Line 515 - No citations, please complete.

Line 523 - No citations, please complete.

Lines 619-620 - Pay attention to correct editing of citations, check throughout the manuscript.

Latin names of mushrooms in references should be all in Italic, please correct!

Author Response

Dear reviewer, we sincerely thank you for your attention in evaluating our proposal and for your valuable contribution to the improvement of our manuscript. We have carefully considered your comments, and the manuscript has been revised accordingly. All the suggested revisions were considered as requested.

Line 77 - He - why isn't there an expanded abbreviation in parentheses, or an explanation of the abbreviation? Please correct.

We thank the reviewer for the suggestion. We modified all He in the text with H. erinaceus.

Line 78 - No citations, please complete.

We thank the reviewer. We added in the text the proper citations where requested.

Line 89-90 - It should add: species and family : „Phylogenetically, Hericium species is classified in the Hericiaceae family within the russuloid linage, very closely to Laxitextum, with steroid sporophore [11].” This should be corrected throughout the manuscript.

We thank the reviewer. We modified the sentence according to the suggestion.

Line 117 - There should be information on whether it is a protected species in danger of extinction. Please add.

We thank the reviewer for the suggestion. We added in the section 2, lines 122-125, a description about that point.

Line 120 - He is an abbreviation, not a Latin name. Remove the italic next to this abbreviation throughout the manuscript.

Done

Line 121 - No citations, please complete.

Done

Line 168 - Citation [34,35] should be at the end of the sentence. Check and correct throughout the manuscript.

Done

Line 261, 290, 293, 386 - No citations, please complete. Each completed paragraph should end with a citation, authors must complete this in many places in the manuscript!!!

Done

Line 263 – Citations should be at the end of the sentence!

Done

Table 4 - The list of abbreviations should be under the table to make it more readable.

We thank the reviewer for the suggestion. We placed the abbreviations as footers under the table .

Line 495 - No citations, please complete.

Done

Line 507 - No citations, please complete.

Done

Line 515 - No citations, please complete.

Done

Line 523 - No citations, please complete.

Done

Lines 619-620 - Pay attention to correct editing of citations, check throughout the manuscript.

Latin names of mushrooms in references should be all in Italic, please correct!

Done.

Reviewer 2 Report

Manuscript Number: jof-2362157

Title: Hericium erinaceus in neurodegenerative diseases: from bench 2 to bedside and beyond. How far from the shoreline?

The manuscript submitted by Federico Brandalise et al. for consideration for publication in“Journal of Fungi ” reviewed available and updated evidences disclosing the neuroprotective and nootropic effects of H. erinaceus dietary supplementation.

Comment 1: line 58, what does “He” mean? The full name should be indicated before the first abbreviation. “H. erinaceus” in Line 80 and “Hericium erinaceus”in line 82, 83, please note the Latin writing and abbreviations throughout the manuscript.

Comment 2: In section 4, line 159-253, “Hericium erinaceus nootropic components” would be clearer if it were described separately in terms of components “erinacines, hericenones, Ergothioneine”. In addition, line 178-182, described bioavailability and tissue distribution, which was related to bioavailability , so line 178-182 should be described in the mechanics section (section 5).

Comment 3: In section 5, “H. erinaceus in vitro, preclinical studies and molecular mechanism underlying the nootropic effects” would be clearer if it were described separately in terms of pathway/ target.

Comment 4: In section 6. “H. erinaceus and its potential on the gut microbiome-brain axis”, mainly described its modulation of the composition of gut microbiota, but lack of the mechanism or ralation of gut microbes/ metabolites and brain/CNS. The section should be added.

Author Response

Dear reviewer, we sincerely thank you for your attention in evaluating our proposal and for your valuable contribution to the improvement of our manuscript. We have carefully considered your comments, and the manuscript has been revised accordingly. All the suggested revisions were considered as requested.

Comment 1: line 58, what does “He” mean? The full name should be indicated before the first abbreviation. “H. erinaceus” in Line 80 and “Hericium erinaceus”in line 82, 83, please note the Latin writing and abbreviations throughout the manuscript.

We thank the reviewer. We modified He” and Hericium erinaceus” in the text with H. erinaceus.

Comment 2: In section 4, line 159-253, “Hericium erinaceus nootropic components” would be clearer if it were described separately in terms of components “erinacines, hericenones, Ergothioneine”. In addition, line 178-182, described bioavailability and tissue distribution, which was related to bioavailability , so line 178-182 should be described in the mechanics section (section 5).

We thank the reviewer. With the purpose to be clearer and more redeable, we have followed the suggestion of the reviewer and we moved some sentences from section 4 to section 5. Furthermore, we described the bioavailability and tissue distribution of erinacines A and S in section 5.

Comment 3: In section 5, “H. erinaceus in vitro, preclinical studies and molecular mechanism underlying the nootropic effects” would be clearer if it were described separately in terms of pathway/ target.

We thank the reviewer. We added a new table, Table 4, summarizing the main in vitro and preclinical studies on H. erinaceus in CNS. Accordingly, we moved some sentences in the manuscript.

Comment 4: In section 6. “H. erinaceus and its potential on the gut microbiome-brain axis”, mainly described its modulation of the composition of gut microbiota, but lack of the mechanism or ralation of gut microbes/ metabolites and brain/CNS. The section should be added.

We thank the reviewer. We added in line 409-418 some sentences about the relation of microbes/ metabolites and brain/CNS. Furthermore, we discussed about a possible direct prebiotic effect and/or immune-modulating effects. Both mechanisms could be involved in gut microbiome composition changes by H. erinaceus.

Round 2

Reviewer 2 Report

1、Section 1 presents the differences between the regulatory systems for H. erinaceus in the East and the West, but less about the East, please add examples of how H. erinaceus is used as a fungal medicine in China or other Asian countries? How to choose the type, what is the dosage, and whether a prescription is required?

2、In the introduction of the new method for Untargeted metabolomic as a new approach for Hericium erinaceus metabolite detection in section 3, the types of metabolites that can be detected by this method and the biochemical properties of Hericium erinaceus metabolites are not elaborated.

3、H. erinaceus can recover and ameliorate pathological brain conditionsincluding Alzheimer's disease,  Parkinson's disease and, spinal cord injury, etc. Why only elaborate on the effects of Hericium erinaceus on depression and cognitive function in sections 7 and 8, and can the effects on other diseases be supplemented? If not, please explain the reason.

4、This article reviews the use of standardized H. erinaceus as a dietary supplement, functional food, or anti-AD drug treatment to promote cutting-edge issues such as brain health. However, throughout section 9 you summarize the research of others too generallyMost importantly, you're simply summarizing other people's research on the role of Hericium erinaceus in neurodegenerative diseases. I have not seen your in-depth thinking about Hericium erinaceus and neurodegenerative diseases, please elaborate and prospect this field.

The section of this article is fluent, the grammar is correct, and the wording is accurate.

Author Response

Dear reviewer, we sincerely thank you for your attention in evaluating our proposal and for your valuable contribution to the improvement of our manuscript. We have carefully considered your comments, and the manuscript has been revised accordingly. All the suggested revisions were considered as requested.

1、Section 1 presents the differences between the regulatory systems for H. erinaceus in the East and the West, but less about the East, please add examples of how H. erinaceus is used as a fungal medicine in China or other Asian countries? How to choose the type, what is the dosage, and whether a prescription is required?

We thank the reviewer for the observation. The sentence regarding the Medicinal Mushrooms (MMs) regulatory systems differences between East and the West is not only related to H. erinaceus, but to all MMs. We underlie this in the manuscript revision. Furthermore, we added in the text the following sentences:

“In China, a number of fungal glycans-based drugs were approved by Chinese Food and Drug Administration (SFDA). Glycans-based drugs approved by SFDA are extracted by hot water either from cultured mycelium and/or sporophore and used clinically in China since 1980s.

Among them, glycans were extracted from sporophore of Lentinula edodes, Polyporus umbellatus, Tremella fuciformis, and from cultured mycelium of Trametes versicolor, Poria cocos, and Grifola frondosa. For example, Krestine (PSK) and PSP, two extracts from T. versicolor, the lentinan extracted from L. edodes, and the schizophyllan from Schizophyllum commune were recognized as drugs in antitumor field, since 1970.

As far as we known, no drugs based on H. erinaceus components are still available for clinical use in China or in Japan. In China, the traditional use of H. erinaceus is mainly based on its gastrointestinal properties as preventive or curative supplement in chronic gastrointestinal disease, such as Crohn’s disease. In Japan traditional use of H. erinaceus is more related to its known neuroprotective properties on central nervous system.”

2、In the introduction of the new method for Untargeted metabolomic as a new approach for Hericium erinaceus metabolite detection in section 3, the types of metabolites that can be detected by this method and the biochemical properties of Hericium erinaceus metabolites are not elaborated.

We thank the reviewer for the suggestion. Accordingly, we implemented the section 3 elaborating the types of metabolites that can be detected by metabolomic and the biochemical properties of H. erinaceus metabolites.

3、H. erinaceus can recover and ameliorate pathological brain conditions,including Alzheimer's disease,  Parkinson's disease and, spinal cord injury, etc. Why only elaborate on the effects of Hericium erinaceus on depression and cognitive function in sections 7 and 8, and can the effects on other diseases be supplemented? If not, please explain the reason.

We thank the reviewer. In the few pilot clinical studies, the effects of H. erinaceus was tested on mild-cognitive impairment patients (Mori et al. 2009), on elderly people (Saitsu et al. 2019), and on mild Alzheimer’s patients (Li et al. 2020). In all these studies, cognitive functions were examined by means of Mini mental state examination (MMSE) before and after H. erinaceus supplementation. Therefore, we preferred to make the section titled “H. erinaceus pilot clinical trials on cognitive functions”.

4、This article reviews the use of standardized H. erinaceus as a dietary supplement, functional food, or anti-AD drug treatment to promote cutting-edge issues such as brain health. However, throughout section 9 you summarize the research of others too generally。Most importantly, you're simply summarizing other people's research on the role of Hericium erinaceus in neurodegenerative diseases. I have not seen your in-depth thinking about Hericium erinaceus and neurodegenerative diseases, please elaborate and prospect this field.

We thank the reviewer for the point. We made the section more personal, explaining our in-depth thinking.